# Skin Cancer Metabolic Profile Assessed by Different Analytical Platforms

**DOI:** 10.3390/ijms24021604

**Published:** 2023-01-13

**Authors:** Yousra A. Hagyousif, Basma M. Sharaf, Ruba A. Zenati, Waseem El-Huneidi, Yasser Bustanji, Eman Abu-Gharbieh, Mohammad A. Y. Alqudah, Alexander D. Giddey, Ahmad Y. Abuhelwa, Karem H. Alzoubi, Nelson C. Soares, Mohammad H. Semreen

**Affiliations:** 1Department of Medicinal Chemistry, College of Pharmacy, University of Sharjah, Sharjah 27272, United Arab Emirates; 2Research Institute of Medical and Health Sciences, University of Sharjah, Sharjah 27272, United Arab Emirates; 3Department of Basic Medical Sciences, College of Medicine, University of Sharjah, Sharjah 27272, United Arab Emirates; 4Department of Basic and Clinical Pharmacology, College of Medicine, University of Sharjah, Sharjah 27272, United Arab Emirates; 5School of Pharmacy, The University of Jordan, Amman 11942, Jordan; 6Department of Clinical Sciences, College of Medicine, University of Sharjah, Sharjah 27272, United Arab Emirates; 7Department of Clinical Pharmacy, Faculty of Pharmacy, Jordan University of Science and Technology, Irbid 22110, Jordan; 8Department of Pharmacy Practice and Pharmacotherapeutics, College of Pharmacy, University of Sharjah, Sharjah 27272, United Arab Emirates; 9Department of Human Genetics, National Institute of Health Doutor Ricardo Jorge (INSA), 1649-016 Lisbon, Portugal

**Keywords:** cancer, skin cancer, omics, metabolomic, metabolic profile, proteomic, biomarkers, analytical platforms

## Abstract

Skin cancer, including malignant melanoma (MM) and keratinocyte carcinoma (KC), historically named non-melanoma skin cancers (NMSC), represents the most common type of cancer among the white skin population. Despite decades of clinical research, the incidence rate of melanoma is increasing globally. Therefore, a better understanding of disease pathogenesis and resistance mechanisms is considered vital to accomplish early diagnosis and satisfactory control. The “Omics” field has recently gained attention, as it can help in identifying and exploring metabolites and metabolic pathways that assist cancer cells in proliferation, which can be further utilized to improve the diagnosis and treatment of skin cancer. Although skin tissues contain diverse metabolic enzymes, it remains challenging to fully characterize these metabolites. Metabolomics is a powerful omics technique that allows us to measure and compare a vast array of metabolites in a biological sample. This technology enables us to study the dermal metabolic effects and get a clear explanation of the pathogenesis of skin diseases. The purpose of this literature review is to illustrate how metabolomics technology can be used to evaluate the metabolic profile of human skin cancer, using a variety of analytical platforms including gas chromatography-mass spectrometry (GC-MS), liquid chromatography-mass spectrometry (LC-MS), and nuclear magnetic resonance (NMR). Data collection has not been based on any analytical method.

## 1. Introduction

Cancer is a public health issue that is considered the second leading cause of death worldwide. It is expected that cancer will replace cardiovascular disease (CVD) as the first cause of death worldwide, in the coming years. However, with the continuous and simultaneous advances in cancer diagnosis and treatment, survivor numbers keep increasing significantly as well. According to the GLOBOCAN estimates, new cancer cases and deaths continued to rise between 2018 and 2020, from 18.1 million [1] to 19.3 million new cases (1.2 million accounting for keratinocyte carcinoma) [2] and 9.6 million deaths [1] to almost 10.0 million deaths from cancer, (0.1 million accounting for keratinocyte carcinoma) [2].

Skin is the largest organ covering the entire human body [3]. It is composed of multiple cell types, including melanocyte cells [4]. Skin cancers are the most common types of cancers with an increasing prevalence globally [5]. A previous study provides strong distinctive evidence which strengthens the hypothesis that climate factors, such as ambient air pollution, global warming, and ozone depletion can lead to an increased rate of skin malignancies globally. It is expected that these factors will continue affecting skin cancer incidence negatively in the coming decades [6]. Early diagnosis and appropriate treatment plans are highly important to achieve survival rates that may reach 99%, depending on the type of skin cancer [4]. Skin cancer is mainly triggered by exposure to ultraviolet rays [3]. However, there is rising evidence that microbiota may also contribute to skin cancer pathogenesis. Although little is known about its role in skin cancer, the recognized association between the knowledge that microbiota modifies the effect of UV- induced immunosuppression, and microbial dysbiosis and inflammation concepts has been explored [5]. Other causes of skin cancer are constant exposure to sunburns and drugs lowering immune system [4].

Skin cancer is classified as melanoma and keratinocyte carcinoma (KC). Melanoma, or malignant melanoma (MM), is considered the riskiest and most lethal type of skin cancer [3]. However, it accounts for only 4% of skin cancer cases, and encompasses 75% of skin cancer-associated deaths because patients are unable to recognize the growth of malignant cells in the initial stages. MM includes various subtypes based on its histopathological features, somatic mutation pattern, and its anatomical distribution. In the case of MM, early diagnosis may be challenging because its morphological characteristics frequently overlap with atypical melanocytic nevi. When MM is suspected, a histopathological examination is routinely performed using a skin biopsy. Although currently considered the gold standard diagnostic for MM diagnosis, it is labor-intensive, time consuming, and can be inconclusive in borderline cases [5].

On the other hand, KC involves all cutaneous malignant neoplasms not related to melanocytes [7]. Including cutaneous squamous cell carcinoma (CSCC), basal cell carcinoma (BCC), cutaneous adnexal tumors, and Merkel cell carcinoma (MCC). KC accounts for 95% of malignant skin cancers [8,9]. CSCC and BCC compromise up to 99% of KCs [7] and are mainly present as localized tumors, not locally advanced or metastatic, and are usually treated with radiotherapy or curative surgery. Furthermore, MCC presents with distant metastasis and is considered one of the most aggressive and rare KCs [8].

## 2. Omics Technologies

“Omics” are branches of science that include the measurements of various disciplines such as metabolites (metabolomics), proteins (proteomics), genes (genomics), or lipids (lipidomics). The term “Omics” is derived from the Latin suffix “ome”, meaning many. The number of these disciplines has increased dramatically with omics studies; however, the number of methodological and biological replicates have not increased in the same manner. Measurements are chemical markers that indicate biological events/endpoints, within the organisms [10].

Recent instrumental analytical approaches greatly impacted biomarker discovery research. The term “biomarker” indicates molecules that are representative of a specific biological state of an organism or cell. Recently, the search for chemical biomarkers for various diseases has become significant, with the routine analysis of biological samples looking for chemical markers beginning in earnest. Nowadays, the analysis of the substantial number of potential biomarkers rather than a few from a single sample has become possible with the methods developed by analytical chemists [11].

The goal of “Omics” studies is to obtain a comprehensive understanding of biological processes in order to recognize the different contributors (e.g., metabolites, proteins, genes, and RNA) rather than each one independently. Today, researchers consider looking for entire classes of biomarkers within an organism, aiming to form associations between their changes/complex interactions and biological events. In the future, the aim of “Omics” research might be to answer more complex questions and become a significant tool in drug development and diagnostics. To date, some complexity of the task is still present [10].

Notably, the medical field is in urgent need of identifying clinically reliable and useful biomarkers to detect cancers at both early and metastatic phases, and is eagerly awaiting biomarkers that can predict which patients with early-stage cancer will benefit the most from adjuvant therapy. In addition, it is critical that such biomarkers are sourced from easily available materials/fluids during the screening and diagnostic steps [11].

MS-based metabolomics and proteomics are promising analysis tools in cancer diagnosis because cancer cells have different metabolic pathways compared to normal healthy cells [12,13]. Metabolism, or metabolic process, is an activity that happens in-vivo, unconsciously. This process produces or consumes small compounds called metabolites. Then, biochemical reactions regulated by proteins and genes in organisms can be described. Recently, the medical field is employing metabolomic studies which are becoming popular methods due to their accuracy to interpret and study the relationship between the pathophysiology of diseases and metabolites.

Metabolomic analysis can be divided into two methods, targeted and non-targeted. In the targeted metabolomic method, biomarkers are pre-determined, and their presence is validated during the analysis. In contrast, the non-targeted metabolomic method is used to detect all metabolites in the sample with more resolution than the targeted method. In cancer metabolomic analysis, several samples can be used to discover cancer biomarkers such as saliva, plasma, urine, and breath [14].Currently, metabolomic analysis is being used to determine cancer biomarkers for diagnostic purposes to discover new targets and to study its complicated heterogeneous nature [15]. Metabolomics can accurately describe an organisms’ current status and reflect accurately what has happened [16].

On the other hand, cancer biomarkers can also be identified using proteomic analysis. Like metabolomic analysis, proteomics can also be divided into targeted and non-targeted methods, which mainly includes protein separation and identification technologies [17]. Researchers study the structure and variations of protein content in cells and organisms in the case of human diseases [18]. Currently, the evolution of Mass Spectrometry (MS) has improved the application of proteomic analysis, especially biomarker discovery. Liquid chromatography with mass spectrometers (LC-MS/MS) is the ideal and most frequently used technique nowadays for proteomics studies, to accurately quantify proteins in biological samples [18,19,20].

In addition to metabolomics and proteomics, genomics is another interdisciplinary field that focuses on function, structure, mapping, evolution, and genome editing. A genome is the complete set of an organism’s DNA, including its three-dimensional structural configuration as well as its genes [21]. Genomic technologies are increasingly used in biomedical research and are starting to have a significant impact on the drug discovery field [22]. They are mainly divided into functional genomics and structural genomics [17]. A few decades ago, scientists were only able to investigate the whole organism’s genome at once instead of only analyzing a small number or even a single gene. Recently, genome or “genomic” science has promptly expanded from a tool to determine DNA sequences to a more functional tool, which studies the roles of both proteins and genes in addition to the expression profiles. Proteomic and genomic techniques are more complex to conduct than metabolomic techniques; several metabolic pathways have been fully studied, thus enabling a better understanding and monitoring of the human body [23].

Moreover, transcriptomics is one of the most advanced fields that study RNA, single-stranded nucleic acid, while a transcriptome is the whole RNA transcripts in a tissue or cell type, under particular physiological conditions, including transfer RNA, messenger RNA, and ribosomal RNA and/or at a certain developmental stage. After genome sequencing, transcriptomics studies the gene expression at the transcriptional level and provides genome information on gene function and gene structure to specify the molecular mechanisms involved in specific biological processes. Then, a regulation network of these biological processes will be revealed and eventually disease diagnosis and clinical therapy will be improved [24]. Metabolites are relatively simple, mostly known, and more suitable for data analysis, due to their larger numbers, compared to genome and proteome. There are up to 40,000 genes in the genome, while in the key metabolic pathways there are only about 3,000 mostly known metabolites [16]. Only limited information can be gathered by conducting a single-omics study. Nowadays, multi-omics studies are critical to getting a more holistic perception and more accurate predictions. Figure 1 summarizes the interaction between the main “Omics” players (metabolome, proteome, genome, and transcriptome) and represents the advantages of metabolomic analysis over other omics [25].

## 3. Skin Metabolites

### 3.1. Skin Metabolites Sources

Analysis of unconventional biological samples, such as skin, has been associated with a significant focus of attention lately, to obtain clinically applicable information, in addition to conventional biofluids such as urine, saliva, and blood.

Interstitial fluid (IF), the fluid filling spaces around cells, provides body cells with nutrients and helps in waste removal. Although its composition is not fully specified, it has been reported by some researchers that it is similar to plasma, due to the presence of lipids and proteins, with lower component concentrations because of their transcapillary flow from the blood to IF. Protein degradation that takes place in the skin’s outer layers, interstitial fluid, sebum, and sweat is the source of low-molecular-weight compounds present within or on the skin. Products secreted from exocrine glands and volatile compounds released by the skin reflect metabolic changes due to cancer progression [26].

Secretory cells of the eccrine gland secrete electrolytes, inorganic ions, amino acids, and elements, across luminal cell membranes as an aqueous fluid, without disintegrating the cells. On the other hand, lipids, proteins, and steroids are secreted as a viscous fluid from the apocrine gland’s secretory cells. Both eccrine and apocrine glands’ secretions contain skin metabolites that can be detected using analytical techniques. Furthermore, other metabolites that are not excreted onto the skin surface can be detected in the epidermis and interstitial fluid. Complete degeneration of glandular cells leads to the holocrine secretion of sebum, which is a complex combination of lipids formed and secreted by sebaceous glands [27].

A wide collection of microorganisms such as bacteria, protozoa, fungi, and viruses represent skin microbiota, which can interact with many components such as topical drugs on the skin, cosmetics, as well as sebaceous, eccrine, and apocrine gland secretions on human skin including volatile organic compounds (VOCs). This interaction may affect the composition of skin metabolites. In addition, VOCs emitted from human skin form odors and provide significant information on body metabolism. Additionally, they are affected by nutritional status, environmental factors, and age [26]. The volatile profile of human skin and body odor changes at an older age, due to decreased hormone levels, lead to decreased secretions from sebaceous and apocrine glands. Other skin metabolites produced from topical drugs such as creams or transdermal patches and cosmetics are available in their outer layers [27].

### 3.2. The Emerging Role of Microbiota and Adipocyte-Derived Lipids

As previously mentioned, there is growing evidence that microbiota may play a role in triggering the incidence of skin cancer. However, microbiota can also affect the treatment response for patients with melanoma [28]. A positive clinical response to treatment in melanoma patients has been associated with specific commensal gut bacteria [29]. Treatment with anti-programmed cell death protein 1 (PD-1) is beneficial for those with metastatic melanoma. The effectiveness of fecal microbiota transplantation (FMT) in combination with anti-PD-1 in melanoma patients was previously studied in clinical study to determine whether changing the gut microbiota may help treat anti-PD-1 resistance. So, in a portion of patients with advanced melanoma, fecal microbiota delivered colonoscopically in combination with anti-PD-1 successfully colonized the responders’ gut and changed the tumor microenvironment, overcoming their initial resistance to anti-PD-1. Additionally, those who responded exhibited higher levels of CD8+ T cell activation, which can fight off cancers caused by altered cells, higher levels of taxa, previously linked to anti-PD-1 responses, and lower levels of myeloid cells, which express interleukin-8, necessary for the survival and proliferation of cancer cells [28].

Immune checkpoint inhibitors (ICIs) have a significant impact on the majority of melanoma patients; nevertheless, those who do not respond to ICIs have fewer options. The effects of human gut microbiota and metabolites on ICI responses in metastatic melanoma patients were examined in a previous study using unbiased shotgun metabolomics, and it was found that ICI responders had high levels of anacardic acid. The stimulation of neutrophils and macrophages by anacardic acids boosts T-cell recruitment to tumor metastases and, thus, increases the ICI. Anacardic acid levels, which have been discovered to have anticancer characteristics, are really influenced by gut flora [30].

On the other hand, adipocyte-derived lipids can influence the diagnosis of melanoma. Through metabolic and lipidomic analysis, researchers studied therapeutic targets for the development of novel antimetastatic medicines, as well as potential biomarkers to predict the metastatic potential of melanoma. Changes in the metabolite and lipid profiles of human melanocytes and melanoma cell lines were found to be associated with cancer metastasis, and metabolomic analysis identified aminomalonic acid as a novel potential biomarker. Furthermore, lipid research revealed a steady increase in phosphatidylinositol (PI) species with saturated and monounsaturated fatty acyl chains with increased melanoma cell metastatic potential, defining these lipids as prospective biomarkers [31].

Despite the usefulness of Adipocytes in the diagnosis of melanoma, it was found in a previous study that they promote the growth and invasion of nearby melanoma cells. Additionally, adipocytes directly transfer lipids to melanoma cells, which modifies the metabolism of tumor cells. The fatty acid transport protein (FATP) lipid transporters that are expressed on the surface of tumor cells convey adipocyte-derived lipids to melanoma cells. Melanomas significantly overexpress FATP1/SLC27A1, one of the six members of the FATP/SLC27A family, and this Melanocyte-specific FATP1 expression works in conjunction with BRAFV600E in transgenic zebrafish to hasten the development of melanoma, an effect also observed in mouse xenograft studies [32].

## 4. Skin or Blood Metabolomic Profiling for Diagnosing and Treating Skin Conditions

Skin metabolomics analysis has a positive impact on disease diagnosis and health improvement. Recently, non-invasive wearable biosensors have been developed as an alternative way for health monitoring, aiming to detect biofluid composition, such as sweat. Metabolites in the interstitial fluid have been detected by the development of slightly invasive microneedle array sensors. The operation steps of the invented biosensor are: electrode collection on a substrate, combining a selective recognition component for the analyte, and applying a physicochemical transducer to transform any sensed variations into legible signals [26]. Skin metabolomic analysis reflects metabolic changes caused by skin conditions, such as skin cancer, while blood metabolomic analysis offers more information about how skin conditions affect the body’s metabolic profile (Figure 2) [27]. The presence of target analytes detected in biofluids is evaluated for any correlation of them in blood using the developed biosensors. However, verifying this correlation is affected by variabilities in sampling protocols, such as the composition and thickness of skin layers (i.e., dermis and epidermis). [26].

In fact, studies have proved that in skin diseases, skin metabolomic profiles are more accurate in following the progression of treatment and reflecting the recovery than blood profiles. As an example, a previous study was performed to examine the metabolic profile of psoriatic skin and blood plasma during treatment with ustekinumab and adalimumab, anti-TNF-α drugs used to treat plaque psoriasis, and to explore the correlation between selected blood and skin metabolites, which are glutamic acid, choline, citrulline, lactic acid, phenylalanine, and urocanic acid, during the treatment, and to compare their levels in both biological matrices. Blood samples were collected from the enrolled patients with time intervals of 1–3 months. Furthermore, for comparison, metabolites present in the psoriatic skin were collected using non-invasive hydrogel micro patch probes that allow the passive diffusion of skin metabolites from the skin to water within the hydrogel matrix.

Study results proved that several metabolites helped monitor treatment response with disease severity and confirmed that skin metabolomic profiles were more reliable in tracing treatment. This explained the reason behind prefering skin metabolites over blood metabolite in skin diseases [27]. However, for most metabolites of clinical interest in other skin diseases, this preference has not yet been confirmed, and blood is still considered the typical biofluid, which has aided in the detection of most human metabolome findings. So, further studies are necessary to find out which is more suitable in the case of other skin conditions [26].

**Figure 2 ijms-24-01604-f002:**
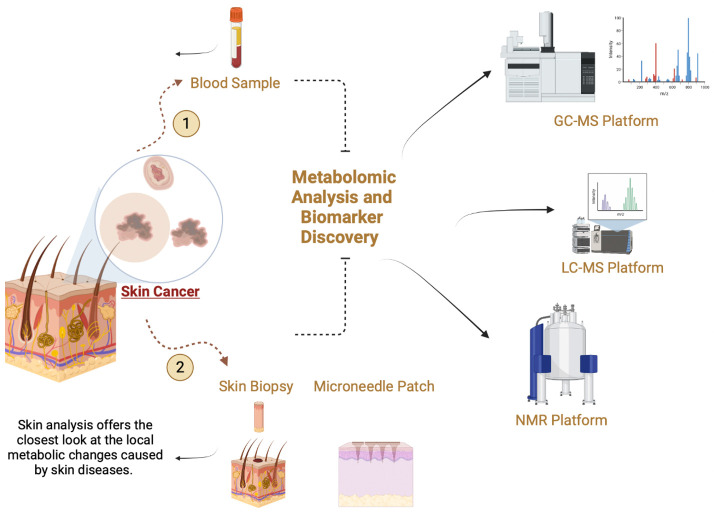
Correlation between Skin and Blood Metabolites.

## 5. Analytical Platform in Metabolomics Workflow

Numerous forms of analytical platforms have been commonly employed in the metabolomics field. However, mass spectrometry and Nuclear Magnetic Resonance (NMR) are the two most popular tools. The latter will be explained later [33].

### 5.1. Mass Spectrometry-Based Metabolomics

Mass spectrometry (MS), a potent analytical tool, has important uses and wide a range of applications nowadays, as it measures the mass of molecules with a high level of sensitivity [34]. It allows the analysis of hundreds to thousands of metabolites on a routine basis [33]. The implementation of Mass spectrometry has taken so long and was not used on a larger scale until the 1990s, because it analyzes gaseous, charged molecules. Although biomolecules are polar and large, they cannot be easily shifted to the ionized gas phase. Currently, Mass spectrometry is also a vital component of the metabolomics and proteomics field [34].

Mass spectrometry-based metabolomics approaches are characterized by their capability to distinguish and quantify thousands of metabolites simultaneously. However, these tasks can be complex and are associated with major challenges, due to metabolites diversity, including dynamic range and chemical complexity. Moreover, metabolite quantification within complex combinations can be difficult due to isomers, ion suppression, and fragmentation. At present, the most comprehensive combinations of analytical methods can detect and quantify around 8000 of the 114,100 metabolites predicted to exist in humans. However, they cannot specify a fixed upper limit for metabolite amounts [35]. In mass spectrometry analysis, metabolites are first exposed to chromatographic separation techniques such as gas chromatography (GC) or liquid chromatography (LC), before detection [33].

In GC-MS and LC-MS-based metabolomic approaches, relative quantification is carried out, which determines fold changes in expression between two different samples, with one being a reference sample, so metabolomic data are detected as relative quantities. In contrast, the NMR based-approach uses absolute quantification, providing absolute concentrations that are directly proportional to peak intensities. The exact amount of the template used for the curve is determined in NMR-based studies.

The first and vital step is the fast enzyme quenching of the metabolic process and terminating all enzyme activities, to get a stable extract of metabolites that quantitively reflects the levels of endogenous metabolite in the original cells and tissues. This reflection is less important in the case of biofluids such as urine, plasma, and serum samples. Indeed, for each tissue type or biofluid, there is a specific method for sampling and extraction. The second step is metabolite separation on a chromatographic column, such as LC, GC, or capillary electrophoresis (CE) columns.

In this review we focus on GC and LC techniques only. The third step is metabolite ionization using an ion source, followed by ion separation in a mass spectrometer, according to their mass-to-charge m/z ratio. Several ionization types are used, such as electrospray ionization (ESI), electron impact (EI), and atmospheric pressure chemical ionization (APCI). Furthermore, mass analyzers used can be time of flight (TOF), Orbitrap, quadrupole (Q), and ion trap (IT). Finally, metabolites are detected and identified through their MS signature and retention time (RT) [35].

### 5.2. NMR-Based Metabolomics

NMR spectroscopy is an additional powerful analytical platform used in metabolomics and proteomics. Although NMR is less sensitive compared to mass spectrometry, it offers many unique benefits to the analysis fields, such as its non-selective nature, high reproducibility, absolute quantitation, and its capability to identify unknown metabolites, as the advances in analytical techniques are helping in the discovery of lots of unknown signals in complex mixtures.

NMR spectroscopy provides data complementary to MS and is often used without any separation methods or sample preprocessing, unlike MS, which is usually combined with GC or LC chromatographic separation techniques to analyze and cover several types of metabolites [36].

Several NMR active atomic nuclei such as ^15^N, ^31^P, ^13^C, and ^1^H can be used in the metabolomic analysis of biological samples. However, ^1^H NMR spectroscopy is the most commonly used because the ^1^H nucleus has a higher NMR sensitivity compared to other nuclei, and is present in nearly all metabolites. Specific metabolites are reliably assigned NMR spectra peaks, with the number of active contributing atomic nuclei being directly proportional to peak intensities. Thus, NMR offers information on metabolite identity and quantity.

As with GC-MS and LC-MS metabolomic workflow, NMR metabolomics also involves several essential steps. Following biospecimen collection such as tissue, urine or blood serum from humans, cell lines, or animal models, metabolite signals can be detected followed by the identification of unknown metabolites by searching the database. Then, a single internal standard can be used for the absolute quantitation of all identified metabolites in the spectrum. After biomarker discovery and validation, metabolite concentrations are used for differentiating between test and control groups, and for identifying changed metabolic pathways, which helps in identifying drug targets to prevent and cure diseases and achieve better therapeutic outcomes [33].

The NMR approach can examine metabolites comprised of many traits considered to be etiologically significant for many diseases, such as triglyceride and cholesterol in the case of coronary heart disease (CHD). However, this approach does not catch other potentially significant factors such as different inflammatory traits, sex hormones, insulin, or regulatory proteins. Therefore, using untargeted mass spectrometry-based metabolomic platforms in addition to NMR could reveal additional metabolites [37]. Figure 3 represents the full process of dermal metabolomics [26].

## 6. Skin Biomarkers Discovery through Metabolomics Applications

Metabolites are the end products of the metabolic process [38]. They are examined in metabolomics research to discover biomarkers, through profiling a specific metabolome followed by their validation for absolute quantification through targeted analysis [26]. Currently, metabolomics can help in monitoring the effects related to a metabolite’s appearance and can be predictive of phenotype in a short time, contrary to gene expression, which takes a few hours for transcriptional changes to disturb downstream pathways [38]. Skin metabolite biomarkers are not used regularly in clinical diagnostics and most biomarker findings are related to urine, tissue, and blood metabolome. Blood biomarkers from serum or plasma are predominantly of clinical relevance [26].

### 6.1. Skin Disease Investigations

Recently, early diagnosis and treatment of skin diseases have been improved by the employment of biomarkers [38]. As mentioned above, even though skin metabolite biomarkers are not routinely used, or used as frequently as other biological sources, many studies of skin biomarkers have been published [38]. For example, in a study that compared sweat samples between healthy control subjects and patients with atopic dermatitis, higher glucose concentrations were found in atopic dermatitis samples. Additionally, glutamic acid, choline, lactic acid, phenylalanine, citrulline, and urocanic acid were revealed as psoriasis-related metabolites in another study [26]. In skin biology, metabolomics has proved useful in the analysis of the effect of both psoriasis and tumors on human skin, basal cell carcinoma, and melanoma [38]. Some metabolites are associated with treatment response, clinical improvement, and disease severity. Skin metabolites are expected to be beneficial biomarkers for more skin conditions and helpful in diagnostic applications. However, there are still inadequate studies available for clinical validation and standardized methods on large representative populations [26].

### 6.2. Sampling Procedures and Analysis of Skin Metabolites

The quality of skin metabolomic analysis is directly influenced by the efficiency of metabolite collection and extraction techniques. Studies on drug metabolism through skin are done in vitro with the employment of skin cells, skin tissue fluid, and skin microsomal and cytoplasmic homogenates, because it is difficult to distinguish skin metabolites from systemic metabolism. On the other hand, studies on drug metabolism through the liver is done in vivo, after the collection of blood or/and urine samples, due to the higher viability of metabolic enzymes in liver, compared to the skin [38].

Recently, various procedures in vivo and in situ have been developed for collecting skin specimens and are used in different studies for skin metabolome and biomarker identification. These procedures are either invasive, minimally invasive, or even non-invasive. Skin is the most exposed, accessible, and largest organ in the human body, and, therefore, there is an interest in the progress of these procedures. Therefore, skin sampling methods have been advanced to address different applications.

Conventional techniques that use matrices from skin layers are utilized to sample skin tissues and perform skin metabolomic analysis, such as superficial fat, dermis, and epidermis, and they need to be obtained by invasive techniques in the clinical setting, such as suction blistering and biopsy [26]. Other minimally or non-invasive techniques are tape stripping [39], iontophoresis [40], micro dialysis [41], ultrasound [42], and microneedle [43]. Table 1 represents several sample collection methods for skin metabolomic analysis [38].

Suction blistering is a less invasive procedure compared to a skin biopsy, and are used in the collection of epidermis and interstitial fluid. Other techniques such as open-flow micro perfusion are used for sampling interstitial fluid in human skin by implanting membrane-free probes with macroscopic openings. In contrast, a skin biopsy is an invasive procedure, and depending on specific approaches, samples can be collected from epidermis, or even from the dermis and superficial fat.

Other non-conventional techniques that collect stratum corneum and skin excretions such as sebum, sweat, and volatile compounds originating from skin are used and considered as non-invasive procedures. A sweat sampling procedure can be done by stimulating the sweat using physical workouts or controlling the environment’s temperature, or by inducing it using iontophoresis and pilocarpine, followed by collection via macro duct sweat collector. However, these inducing methods may affect the sample composition. Then, the skin surface can be washed with extraction buffer or solvent to collect sweat and sebum. In addition, they can be collected without contacting the skin, by using a solid-phase microextraction device, or by using different sorbent materials designed within a patch, such as polydimethylsiloxanes and hydrogels that directly contact the skin [26].

## 7. Skin Cancer Metabolomics Using Different Analytical Platforms

### 7.1. GC-MS Based Skin Cancer Metabolomics

Among several analytical techniques used in the metabolomic analysis, GC-MS based metabolomics is the most spectral reproducible, and allows the measurement of volatile/semi-volatile thermally stable metabolites [44]. Many studies confirmed and linked the strong olfactory capability of dogs to distinguish and detect cancer from non-cancer samples to the presence of volatile molecules in the headspace over melanoma [40,41,42].

The analysis of volatile organic compounds (VOC) as cancer biomarkers is both challenging and promising and is considered a novel diagnostic approach to detect cancer by metabolomic analysis of breath and urine, as well as skin [45]. R. Anthony DeFazio and his group expanded their efforts to achieve an early melanoma detection by identifying melanoma volatile metabolic signatures using untargeted GC-MS based metabolomic analysis. Forty-nine male melanoma patients were involved in this study, and samples were obtained from fresh melanoma tissues and nearby non-neoplastic healthy skin tissues, collected from the same patient, to be used as control. Firstly, an excisional biopsy sampling method was used to collect the samples using 2-mm punch device, one month after melanoma diagnosis was obtained, followed by the collection of volatile compounds using the Head Space Solid Phase Micro-Extraction method (HS-SPME) [46]. This method of VOC collection above cancerous tissues will probably become a non-invasive in situ collection without the need of biopsies in the future, thus speeding up biomarker discovery and validation. GC-MS was used to separate and analyze volatile compounds. However, many of these compounds could not be identified due to sample complexity. In summary, this study identified 32 volatile compounds, 9 of which showed 100% enrichment in melanoma samples compared to healthy skin tissues (Table 2), and 23 were distinguished only in melanoma and not in normal tissues (Table 3).

In addition to volatile compound collection and analysis, histologic analysis was performed for melanoma and healthy skin samples using hematoxylin/eosin staining. Melanoma samples were positive for malignant melanoma with 0.78 Breslow thickness, and non-neoplastic healthy skin showed signs of solar elastosis [45], which is a degenerative condition associated with chronic damage of the elastic tissue in the dermis layer due to extended sun exposure [47].

Examples of volatile organic compounds found only in melanoma are 1-hexadecene, isopropyl palmitate, ethylene oxide, Bis (2-ethylhexyl) phthalate, amide formamide, and undecane. Increased metabolism and fatty acid synthesis in melanoma progression led to the formation of palmitic acid derivatives, such as 1-hexadecene and isopropyl palmitate. Moreover, ethylene oxide was detected only in melanoma samples although there are no data that support the formation of endogenous ethylene in animals or humans. 1-aminocyclopropane-1-carboxylate synthase (ACC synthase) is a human enzyme responsible for ethylene synthesis, however, it is allegedly inactive due to mutations of two amino-acid residues within the enzyme’s binding site. Thus, it seems that ethylene oxide in melanoma samples is not a primary metabolite. Though intestinal bacteria are supposed to increase ethylene levels, the precursor of ethylene oxide via metabolism and lipid peroxidation, additional studies are needed to explore this mechanism in human skin. The presence of ethylene oxide due to contamination in the punch biopsy device is ruled out because it was not detected in the normal skin samples.

Bis(2-ethylhexyl) phthalate has also been found only in melanoma and not in healthy samples. In general, anthracyclines anticancer drugs are metabolized to phthalic acids through oxidative degeneration, with phthalate esters commonly used as plasticizers. However, our patient did not receive any anthracyclines [46]. Bis (2-ethylhexyl) phthalate was previously identified to be elevated in melanoma samples compared to both nevi and normal skin [48], so possibilities of contamination and anthracyclines anticancer drugs can be excluded.

On the other hand, other volatile organic compounds have been found in both samples but showed 100% enrichment in melanoma samples, compared to healthy skin samples such as 1-hexadecanol, Methyled benzene (Benzene, 1,3,5-trimethyl), and nonanal. 1-Hexadecanol showed a 35-fold increase in melanoma samples, as compared to healthy skin samples, which may reflect the increased synthesis of de novo fatty acid (FA), which is a critical metabolic modification in cancer cells used to synthesize new plasma membrane, considering 1-Hexadecanol as a potential biomarker of melanoma [46].

In another study, R. A. DeFazio and his co-workers conducted an untargeted approach to detect and compare volatile metabolomic signatures of melanoma sample (M) and keratinocyte carcinoma (KC) samples, using healthy adjacent skin as a control from the same patient to be collected, and analyzed on the same day. Non-melanoma cases were confirmed histologically after testing excisional biopsy samples and assured the absence of residual melanoma. Healthy adjacent non-neoplastic skin is the optimal control because there are differences in lipids and their secondary metabolites due to cancer progression, in addition to age and topographic related differences in the lipid structure of human skin.

Like the previous study, VOCs from biopsied samples were collected using a HS-SPME device with a 0.65 um Polydimethylsiloxane/Divinylbenzene (PDMS-DVB) fiber at room temperature for one hour, and analyzed using GC-MS: however, in this study, the number of recruited patients were 10. Five patients had melanoma tumors and five were recruited with benign tumors or keratinocyte carcinoma. A 2 mm punch biopsy device was used to collect two tissue biopsies from each patient, one from a melanoma or keratinocyte carcinoma lesion and the other from healthy adjacent skin. A total of 20 tissues were analyzed, five keratinocyte carcinoma samples versus five healthy adjacent skin, and five melanomas versus five healthy adjacent skins. In addition, 10 air samples from the empty vial containing the volatile collection were analyzed. Comparison of the metabolomic signatures between three groups: skin vs. air samples, keratinocyte carcinoma vs. matching skin, and melanoma vs. matching skin is shown in (Table 4). Volatile compounds were annotated based on their retention index (RI) and spectral similarity with a library match.

Palmitic and lauric acid are revealed to be significantly elevated in melanoma samples, relative to the control skin. They are listed in the Human Metabolome Database as one of the most commonly found saturated fatty acids in animals with measurable concentrations in human urine, cerebrospinal fluid, and serum [45]. Plasmalogens are essential membrane glycerophospholipids with vinyl ether linkages that present normally in the plasma membrane at an amount of no more than 1% [49]. However, more than 4% of the plasmalogens of total phospholipids indicate neoplastic transformation. The increased amount of palmitic acid in the melanoma group is highly likely related to its function as a building block for plasmalogens synthesis. Moreover, cell proliferation in cancer leads to increased lipid synthesis, which is supported by palmitic acid presence. In addition, lauric acid, a medium chain fatty acid, can be formed from increased plasma membrane lipid peroxidation and increased aldehyde dehydrogenase (ALDH) activity in many cancers, including skin cancer. It is difficult to be confident about cinnamaldehyde origin in melanoma samples because they are known to be present in cosmetics as well as skin.

However, toluene was the only volatile compound found significantly decreased in melanoma samples relative to the skin controls. It is assumed that toluene is not an endogenous metabolite, but is an environmental contaminant although it was identified in a few studies as a down regulated compound in lung cancer patients (non-smokers). It is certain that none of these identified volatile compounds are secondary metabolites of the medications used by melanoma patients, because they were identified relative to the paired healthy control skin from the same patient [45]. Finally, for the future of metabolomic biomarker discovery in skin cancer using GC-MS platform specifically, a larger number of cases is required with larger efforts to fully validate these findings and assign reliable biomarkers.

**Table 4 ijms-24-01604-t004:** Comparison of metabolomic signature between three groups: skin vs. air samples, keratinocyte carcinoma vs. matching skin and melanoma vs. matching skin.

Skin vs. Air Samples	Non-Melanoma vs. Matching Skin	Melanoma vs. Matching Skin
10 air samples were analyzed against 10 skin samples (five keratinocyte carcinoma and five melanoma) Volatile components in these two group samples did not match, meaning that they were collected at different times Identified metabolic features identified as significant with P value less than 0.05 were 158.	5 keratinocyte carcinoma samples were analyzed against adjacent matching control skin from the same patient. No dysregulated metabolic features were identified between the two groups Melanoma is absent in this group consequently histologic analysis did not discover any neoplastic changes. So, the incapability to discover any differentially expressed features between adjacent control skin and keratinocyte carcinoma is expected.	5 fresh confirmed melanoma samples were analyzed against adjacent matching control skin from the same patient Annotated volatile compounds identified to be significantly increased in melanoma samples relative to control skin were; 2-Ethylhexyl-4-methoxycinnamate, 1-eicosene, palmitic, lauric, and myristic acid. Only one volatile compound, toluene was found to be decreased in melanoma samples relative to control skin.

### 7.2. LC-MS Based Skin Cancer Metabolomics

Liquid chromatography-mass spectrometry (LC-MS) is another platform utilized by metabolomics. This technology combines the mass spectrometer’s mass analysis abilities with the liquid chromatography’s physical separation. An ion source, a mass analyzer, and a detector are the three main components of a mass spectrometer. Sample molecules are transformed into ions by the ion source, which are then resolved by the mass analyzer, either in an electromagnetic field or a time-of-flight tube, before they are assessed by the detector. Ion sources come in various forms, such as electrospray ionization (ESI), atmospheric pressure photoionization (APPI), and atmospheric pressure chemical ionization (APCI). To enhance metabolome coverage, it is sometimes necessary to examine the biological sample in both positive and negative ionization modes, using a scan range of m/z 50–1000 [50]. Significant advancements in chromatographic resolution and accelerated analysis have also been made recently with the help of ultra-performance liquid chromatography (UPLC), which is characterized by a pressure > 400 bar in columns packed with 2-m diameter particles [51]. For improved attribution of metabolites from chromatographic mass data, UPLC and MS are often hyphenated.

To study the molecular changes between primary cutaneous melanoma and the nearby normal skin, Taylor et al. conducted a metabolomics analysis for primary melanoma and matched extra-tumoral microenvironment (EM) tissues, in order to widen temporal inferences related to the commencement of melanoma, and compared mismatched metastatic melanoma tissues to EM tissues to predict its progression. Utilizing a m/z of 70 to 1000, which corresponds to the mass range of most predicted metabolites, metabolic profiling was performed using UPLC on frozen human tissues, which revealed significantly different levels of metabolites in the melanoma tissue, when primary or metastatic melanoma was compared to EM (75% and 68%), respectively. In contrast, there were no significant variations between the abundances of the metabolites in primary and metastatic melanoma.

Pathway-based analysis revealed three significantly important pathways between primary melanoma and EM; ascorbate and aldarate metabolism, propanoate metabolism, and tryptophan metabolism, as well as three significant pathways between metastatic melanoma and EM; ascorbate and aldarate metabolism, pyrimidine metabolism, and histidine metabolism. Most individual metabolite abundances found within these pathways were directionally consistent, such that the majority of compounds found to be more or less abundant in primary melanoma compared to EM were also more or less abundant in metastatic melanoma compared to EM.

The metabolites in the aldarate and ascorbate metabolic pathway set the primary and metastatic melanoma’s metabolomes apart from those of EM. The metabolic pathways for tryptophan and propanoate separated initial melanoma from EM but not metastatic melanoma from EM, indicating that these pathways may be crucial for the carcinogenesis of melanoma. The metabolic pathways for pyrimidine and histidine, however, were unable to differentiate primary melanoma from EM. This data suggests that rather than its initiation, these pathways may be crucial to the progression of melanoma, and the proteins or metabolites in these pathways may function as possible treatment targets to prevent metastasis [52].

In another study, the effectiveness of the monoclonal anti-programmed-death-1 (PD-1) antibody drug, Nivolumab [53], was examined in research including 82 patients with metastatic renal cell carcinoma (RCC) and 79 patients with advanced melanoma [54]. The authors discussed the safety and effectiveness of nivolumab in several malignancies to date, including advanced melanoma, renal cell carcinoma (RCC), and non-small cell lung cancer (NSCLC). Tumor cells were hypothesized to manipulate signals that decrease T-cell responses, in the evasion of the immune system. Both PD-1 on the T cell and its ligand PD ligand1 (PD-L1) on the tumor cell interact in a significant way to cause this event. Nivolumab and other PD-1 pathway inhibitors are consequently able to reverse T-cell suppression and ultimately trigger antitumor responses [53]. On pre- and post-treatment blood samples from 79 patients with advanced melanoma and 82 patients with metastatic RCC receiving nivolumab, liquid chromatography-mass spectrometry was carried out. More than 100 identified metabolites were accurately assessed at baseline (before nivolumab was started), and then at different time points (4, 6, and 9 weeks), following treatment commencement, and were related to the greatest overall response as well as progression-free survival (PFS). The findings showed that the change in kynurenine levels in melanoma patients was strongly associated with the response to nivolumab. Additionally, worse progression-free survival (PFS) and lack of response to nivolumab were related with greater baseline levels of adenosine in RCC patients. Briefly, a correlation between the change in kynurenine levels and response in melanoma patients was observed. Additionally, worse PFS and lack of response to nivolumab are linked to greater baseline levels of adenosine in RCC patients. However, it was implied that serum metabolites could have a potential role as indicators of PD1 inhibition [54].

### 7.3. NMR Based Skin Cancer Metabolomics

In addition to GC-MS and LC-MS, NMR spectroscopy is another technique widely recognized for revealing the structure of molecules. The quantitative link between intra-molecular and inter-molecular resonances is revealed by the ^1^H (proton) NMR spectrum with chemical shift and coupling constants [55]. For the collection of metabolomics data, NMR has many benefits. It may often be used in a non-invasive way, requiring little to no sample preparation, and is non-destructive. It is suitable for research on biofluids, cell extracts, and in vitro or in vivo cell cultures and tissues. The NMR’s multinuclear capabilities offer a variety of ways to study diverse compounds [56]. Table 5 compares the three different analytical platforms.

Being the deadliest form of skin cancer, melanoma causes metabolic rewiring and intensifies metastatic transformation. Preclinical models, particularly cell lines, are a major part of efforts to enhance its early and accurate diagnosis. To map the metabolic changes presumably regulating metastasis, a study set out to describe a combinational NMR and ultra-high-performance liquid chromatography-high resolution tandem mass spectrometry (UHPLC-HRMS/MS)-mediated untargeted metabolomic profiling of melanoma cells, by looking at two patient cell lines, WM115 and WM2664. WM115 was obtained from a primary, pre-metastatic tumor, while WM2664 was clonally amplified from lymph node metastases. NMR and UHPLC-HRMS were used to examine metabolite sample data.

The metabolites were identified by multivariate (S-plot, Variable Importance in Projection (VIPs)) and univariate statistical analysis. Metabolites that arose from both statistical methods include lactic acid, myo-inositol, glutamic acid, creatine, succinic acid, proline, o-phosphocholine, and choline. These may act as molecular biomarkers for human advanced melanoma that is metastatic, and harbor great prognostic, diagnostic, and therapeutic potential [62]. The upregulation of glycolysis and the enhanced synthesis of lactic acid is one of the most significant changes found in this study due to the Warburg effect. To support growth, survival, proliferation, and long-term maintenance, cancer cells rearrange their metabolism. The increased absorption of glucose and the fermentation of glucose to lactate are frequent characteristics of this altered metabolism. The Warburg Effect refers to this phenomenon, which may be seen even when the mitochondria are fully functional [63].

Myo-inositol was another significantly altered metabolite. In-vitro and in-vivo NMR investigations reveal an association between myo-inositol’s involvement in controlling osmotic pressure and tumor-cell density [64]. Moreover, the metabolite hypoxanthine, which was found to be at least 14-fold greater in WM2664 than in WM115 cells, is derived from the purine biosynthesis pathway [65]. Since this pathway is characterized by the transformation of ribose 5-phosphate into inosine monophosphate, which is subsequently dephosphorylated to create inosine, the main precursor molecule for hypoxanthine, the drastically decreased inosine contents in the WM2664 metastatic melanoma cells were seen to be tightly correlated with the noticeably raised levels of hypoxanthine. In the primary melanoma WM115 cell group, the important cell membrane constituents choline, phosphocholine, and glycerophosphocholines also showed a statistically significant increase. This could be attributed to the irregularities in choline metabolism that have been closely linked to abnormal cellular membrane production and cancer.

In light of everything, the UHPLC-HRMS analysis revealed significant differences in 30 out of the 36 metabolites between the two corresponding metastatic phases. Differently produced, secreted, and/or sequestered metabolites are part of a variety of key metabolic processes, including the catabolism of glucose, glutamine, and choline, which are essential for the survival and expansion of advanced melanoma cells. Through the utilization of this platform, the study was able to shed light on the molecular biology of advanced melanoma and open the possibilities of new treatment avenues for the metastatic disease form(s) in the clinic [62].

^1^H NMR-based metabolomics platform was used in another study to investigate the metabolic variations of squamous cell carcinoma (SCC), peri-tumoral zones, and distant normal skin. A total of 24 patients with confirmed diagnosis of SCC were included and tissue samples were collected from the three regions from each patient. Samples collected from distant normal skin (> 5 mm adjacent to the tumor lesion) were used as controls, and peri-tumoral zones samples were considered as the tumor “bed”. ^1^H NMR spectroscopy was carried out and the results showed that 186 metabolites were identified and quantified. Table 6 summarizes the findings of the study.

The study findings revealed that amino acid metabolism, glycolysis, and the tricarboxylic acid (TCA) cycle had a vital part in the metabolism of SCC and peri-tumoral areas [66]. One of the essential features of cancer is the re-adjusting of energy metabolism to meet the biosynthetic and bioenergetics requirements for the growth and proliferation of anabolic tumors [67]. Higher rates of glucose consumption and preferential secretion of lactate were observed in the proliferative tumor cells relative to non-transformed cells, to support the required pools of acetyl-CoA and lipogenesis, known as the Warburg effect. In addition, glutamine is considered a major fuel for tumor cell proliferation by supporting the glutaminolysis pathway [66]. In addition, the significant increase in glutamine levels in SCCs and peri-tumoral regions may support its continuous flux into the TCA cycle and ensure nucleotide and amino acid synthesis [68]. Unlike normal skin, the glutaminolysis pathway was more dominant in peri-tumoral regions than the glycolysis pathway, in supplying the required energy and maintain their TCA cycle [66]. As a result of these findings, further studies of the biochemical pathways that trigger metabolic activity differences between SCC and peri-tumoral regions may be helpful to discover new therapeutic targets to diagnose and treat SCC in its early stages.

## 8. Clinical Implications of the Metabolic Profile of Skin Cancer

There is a wide range of approaches for treating skin cancer, including excision, radiation therapy, cryosurgery, electrodesiccation, and curettage and immunotherapy (PD-1 inhibitors) [69]. However, treatment may often result in undesired long-term side effects, like fatigue and increased risk of cardiovascular diseases and cardiotoxicity [70]. Importantly in some cases, tumor cells develop resistance to therapy representing a major drawback in the success of skin cancer treatment [71]. The Food and Drug Administration (FDA) has approved two PD-1 inhibitors, cemiplimab and pembrolizumab, as systemic therapies for recurrent or metastatic skin cancer. Each of these treatments not only has a low success rate, but also has associated side effects such as toxicity and death [72]. As a result, the biomedical community must urgently create reliable and precise biomarker profiles to assess and monitor therapy responses in skin cancer patients [73]. The metabolic profile of skin cancer has several clinical implications, such as improving the clinical practice using disease metabolomics data by clinicians as a diagnostic, prognostic and monitoring tool for skin cancer, and confirming to clinicians how successful treatments are for the management of patients with skin cancer by tracking patients’ therapy responses routinely. Clinical impacts also involve showing molecular basis of cancer development and progression in addition to identifying new candidate biomarkers, allowing for the development of more effective, tailored anticancer therapy.

Although metabolomics can be used to track a treatment, metabolite levels on their own provide a snapshot of the metabolism at the time of collection. To anticipate the result of skin cancer immunotherapy and provide information on treatment efficacy, while tracking toxicity levels, it is therefore required to supplement this information with metabolic pathway dysregulation and other omics changes. However, melanoma is a remarkably heterogeneous tumor type that significantly varies from patient to patient. In an effort to reduce the percentage of patients who do not respond to treatment, precision medicine has paved the way for increasingly tailored treatments, in order to provide each patient with the most suitable care and to direct treatment choices. Precision medicine has proven a success at enhancing disease outcomes, improving quality of life, and discovering and overcoming drug resistance and relapse mechanisms, while treating patients with melanoma. It can predict patient responses to medicines or probable adverse events by developing new predictive and/or prognostic biomarkers. In this scenario, multi-omics analyses (genomics, transcriptomics, proteomics, and metabolomics) are used to support the clinical management of patients with melanoma [74].

## 9. Limitations

Nowadays, omics studies are considered a rapidly advancing suite of tools for clinical research that enable us to gather and analyze biological samples in increasingly novel ways [60,61]. Despite the different omics applications presented in this review, omics techniques retain some limitations. Firstly, untargeted metabolomics is a hypothesis generating approach, it focuses on gathering data of small molecules in a biological sample that correlate with a specific heath condition, without having any previous knowledge of metabolite identity. However, this approach is acceptable if the goal is to generate hypotheses about metabolic pathways that lead to different responses to a disease [58,59]. Due to the differences between several parts of the skin, analyzing skin metabolome is challenging. The challenges to address in skin metabolomics are many. Firstly, skin is a dynamic medium; therefore, metabolomics cannot comprehensively capture dynamic changes in all metabolites as their levels may change over time [38].

In addition, interpersonal variability of metabolite levels is a main concern, which can be a result of non-uniform metabolite distribution on the skin surface. In addition, the small number of biological samples collected, and low analyte concentrations cause difficulty in determining an absolute concentration [75,76]. Moreover, the reaction of the skin to different stressors affecting it may vary between individuals, regardless of whether they belong to a healthy or diseased group in a study [26].

## 10. Conclusion and Future Perspective

Recently, there has been a huge growth in the practice of “Omics” based approaches in medical research. It is now possible to produce a large amount of “Omics” data in a time- and cost-effective manner by the advancements in laboratory protocols and data storage. Nowadays, “Omics” approaches are getting us closer to the clinical phenotype of complex diseases that may involve multiple genes, organs, and are affected by environmental factors. “Omics” approaches are ultimately mapping the entire cellular pathways that help in conducting in vivo studies. Our knowledge of human diseases is expanding with “Omics” studies, particularly metabolomics, aiming to explore a new generation of biomarkers. The future of skin metabolomics as an effective research approach depends on developing reliable and convenient sampling methodologies, sample preparation and biomarkers detection, and data analysis and validation. Also, skin metabolomics is expected to enhance the advance of “precision medicine” in the dermatology field.

## Figures and Tables

**Figure 1 ijms-24-01604-f001:**
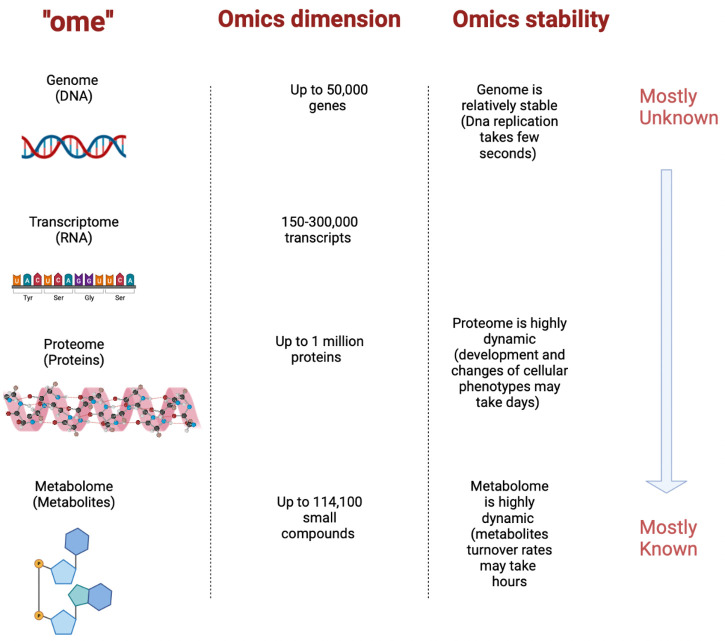
Metabolomic analysis advantages over other omics technologies.

**Figure 3 ijms-24-01604-f003:**
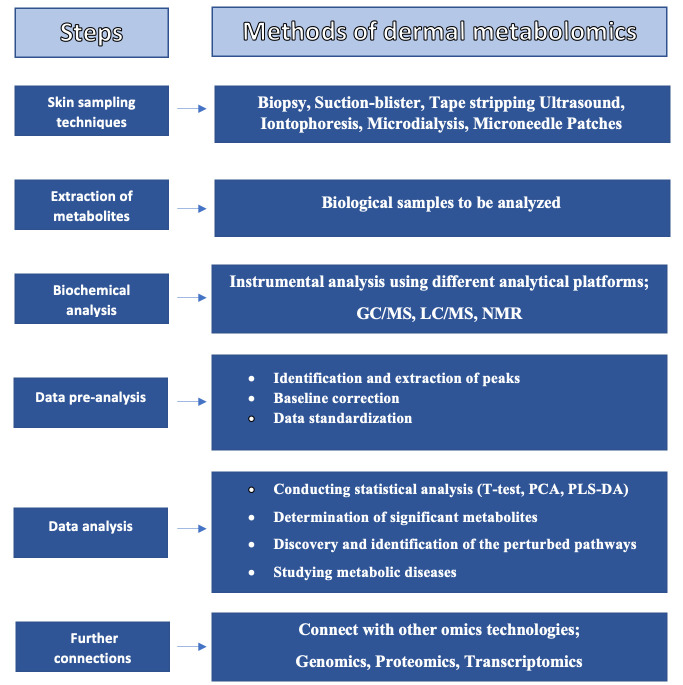
Process of dermal metabolomics.

**Table 1 ijms-24-01604-t001:** Several sample collection methods for skin metabolomic analysis.

Sampling Method	Advantages	Disadvantages
Biopsy [38]	Specimens can be collected from or beneath the epidermis (including superficial fat and dermis)	Patient unfriendly (invasive technique) Time-consuming
Suction-blister [38]	Less invasive compared to skin biopsy Can be used to collect the epidermis and interstitial fluid	Time consuming Specialized equipment required
Tape stripping [39]	Minimally invasive technique Collect samples from the stratum corneum	Sample preparation often needs to be completed before metabolite analysis such as sample clean-up, extraction, and analyte preconcentration. Lack of standardized protocols
Ultrasound [42]	Efficient extraction	Possible denaturation of the metabolite Many impurities
Iontophoresis [40]	Minimally invasive technique	Local stimulation (partial penetration of the skin)
Microdialysis [41]	Minimally invasive technique Can also be used for sampling interstitial fluid in the skin by implanting thin tubular membranes	Time consuming Limited to small molecules
Microneedle Patches [43]	Minimally invasive technique Helpful in collecting skin interstitial fluid	Limited extraction efficiency

**Table 2 ijms-24-01604-t002:** Volatile compounds that showed ≥ 100% enrichment in melanoma compared to the healthy skin sample with the % Total Ion Count, (%TIC) and their ratios in Melanoma vs Skin and M/S ratios.

Volatile compounds	%TIC in Melanoma (M)	%TIC in Healthy Skin (S)	RATIO M/S
**1,2-Benzenedicarboxylic acid, diisooctyl ester**	1.03	0.61	1.68
**1-Hexadecanol**	3.29	0.09	35.90
**2-Ethylhexyl trans-4-methoxycinnamate**	1.02	0.78	1.31
**Benzene, 1,3,5-trimethyl**	0.12	0.03	3.41
**Dichlorodifluoromethane**	0.03	0.01	1.80
**Dodecane**	0.13	0.11	1.27
**N-Morpholinomethyl-isopropyl-sulfide**	0.32	0.23	1.37
**Nonanal**	0.30	0.11	2.79
**Undecane, 4,7-dimethyl**	0.17	0.11	1.54

**Table 3 ijms-24-01604-t003:** 23 Volatile compounds found only in melanoma.

Compounds Found Only in Melanoma
1,3-Cyclopentadiene, 5-(1-methylethylidene)-
1-Eicosanol
1-Hexadecene
2-Ethoxyethyl acrylate
2-Isopropylamino-4-methylbenzonitrile
Adenosine, 5’-amino-5’-deoxy
Benzaldehyde, 4-methoxy
Bis(2-ethylhexyl) phthalate
Cyclohexane, ethyl
Decanal
Decane, 2-methyl
Decane, 4-methyl
Ethanol, 2-butoxy
Ethylene oxide
Formamide
Heptane, 2,4-dimethyl
Isopropyl Palmitate
Nonane, 1-iodo
Pentane, 2,3,4-trimethyl
Phthalic acid, isobutyl 4-octyl ester
Spiro [bicyclo [2.2.1] hept-5-ene-2,1’-cyclopropane]
Undecane

**Table 5 ijms-24-01604-t005:** Comparison between different analytical platforms.

Analytical Platform	Advantages	Disadvantages	Detectable Molecules
GC-MS [57,58]	Sensitive platform to detect a wide range of metabolites Reproducible technique. Excellent separation Quantitative technique Compatible for gases and liquids Able to detect most organic and some inorganic compounds Suitable for analysis of mixtures Generally cheaper than LC-MS because it uses a less sophisticated detector.	Destructive technique Limited to volatile and thermally stable compounds Cannot analyze solids Derivatization required Separation required Long analysis time (20–40 min/sample)	Organic acids Amino acids Fatty acids Glucose Amines Polyols
LC-MS [58,59]	Highly sensitive technique Derivatization is not required Can analyze thermally unstable compounds Compatible with liquids and solids Small sample volumes are needed	Destructive technique Expensive instrument and needs skilled personnel Ionization suppression resulting from co-elution of molecules with dramatic variances in concentrations Unwanted solvent matrix effects Long analysis time (15–40 min/sample) Not suitable for gases	Organic acids Nucleotides Active peptides Vitamins Bile acids Flavonoids Polyphenols Lipids Carbohydrates Amino acids
NMR [43,60,61]	Non-destructive technique. The sample remains intact after the analysis. Quantitative technique Single internal reference is enough for absolute quantitation of all metabolites in the spectrum. Does not involve harsh sample treatment before or during the analysis. Enables the analysis of bio-fluids and tissues without the need for sample separation and preparation. Short analysis time (2–3 min/sample) Can analyze liquids and solids. No derivatization needed. Automated technique.	Low sensitivity. Expensive instrument. Unable to detect/identify inorganic ions and salts. Unable to detect non-protonated samples. Requires large sample volumes (0.1- 0.5 mL)	Most molecules

Abbreviations: GC-MS, gas chromatography-mass spectrometry; LC-MS, liquid chromatography-mass spectrometry; NMR, nuclear magnetic resonance.

**Table 6 ijms-24-01604-t006:** Study findings.

Groups	Metabolomics Signatures
I Between SCCs and controls	Eight metabolites were recognized and validated. The concentrations of glutamine, tyrosine, aspartate, creatinine, valine, alanine and lactate were significantly increased in SCCs group compared to controls. The concentration of glucose was significantly decreased in SCCs group compared to controls.
II Between peri-tumoral regions and controls	Four metabolites were identified and validated. The concentrations of aspartate, valine, lactate and alanine were significantly elevated in peri-tumoral regions compared to controls.
III Between peri-tumoral regions and SCCs	Four metabolites were identified and validated. The concentrations of aspartate and phenylalanine were significantly elevated in peri-tumoral regions compared to SCCs The concentrations of isoleucine and methanol were significantly diminished in peri-tumoral regions compared to SCCs.
IV Between the three groups	11 metabolites (methanol, glutamine, tyrosine, aspartate, phenylalanine, creatinine, alanine, lactate, isoleucine, valine, and glucose) were identified to be significant among all the three groups. 3 of them (valine, lactate, and alanine) involved in peri-tumoral zones and SCCs were recognized as the main affected metabolites.

## Data Availability

Not applicable.

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
