# Peer review of "Skin Cancer Metabolic Profile Assessed by Different Analytical Platforms"

_ijms, 2023, doi:10.3390/ijms24021604_

Round 1
Reviewer 1 Report
Thank you for the opportunity of reviewing this article. It is interesting and quite well written.
My remarks are as followes:
11. Currently the term non-melanoma skin cancers is out of use, such cancers should be named keratinocyte carcinoma (KC) (see and cite: Detailed head localization and incidence of skin cancers. Fijałkowska M, Koziej M, Antoszewski B. Sci Rep. 2021 Jun 11;11(1):12391. doi: 10.1038/s41598-021-91942-5.)
22. In the sentence „They are one of the most common types of cancers with an increasing prevalence globally [5]” please remove „one of” as skin cancers are the most commont types of cancers!
33. In the sentence „Early diagnosis and appropriate treatment plans are highly important to achieve survival rate that may reach 99% [4]” please add „depending on the type of skin cancer”. Very good prognosis are in BCC but totally diferent situation is in SCC or MM. And in the next paragraph Authors are writting about MM and possible 75% rate of deaths so some inconsistency in thinking is present…
44. Paragraph from line 85 about NMSC (change to KC) needs revision, read above mentioned article „Detailed head localization and incidence of skin cancers”. In the sentence „It accounts for 30% of all cancers, with a low overall mortality rate despite high occurrence, contrary to MM [8].” It at the beggining of sentence is for BCC, CSCC or NMSC?
55. In point 4 Correlation between Skin and Blood Metabolites remove „subcutaneous” from skin layers!!!! Subcutaneous tissue is the layer under the skin but totally not the part of it. Skin has 2 layers: epidermis and dermis. I do not understand the connection between this paragraph and skin cancers, the Authors write here about psoriasis, glucose level, but there are no correlations between skin and metabolites in skin cancers?!
66. Similar situation in in points 6, 7, and 8. Maybe points from 4 to 8 should be presented together as the examples and as some kind of introduction to similar connection in skin cancers. Think of reorganizing.
77. What is the take home message from your article especially for clinicians? In what way presented knowlegde can be „translated” to management in patients with skin cancer?
Author Response
Point 1: Currently the term non-melanoma skin cancers is out of use, such cancers should be named keratinocyte carcinoma (KC) (see and cite: Detailed head localization and incidence of skin cancers. Fijałkowska M, Koziej M, Antoszewski B. Sci Rep. 2021 Jun 11;11(1):12391. doi: 10.1038/s41598-021-91942-5.) This summary is accurate, and we thank the reviewer for their time and effort in reviewing this manuscript.
- Response 1: We thank the reviewer for the suggestion, the term has been changed as required to keratinocyte carcinoma (KC) in all the manuscript
- Point 2: In the sentence „They are one of the most common types of cancers with an increasing prevalence globally [5]” please remove „one of” as skin cancers are the most common types of cancers!
- Response 2: This has been removed, as required.
- Point 3: In the sentence „Early diagnosis and appropriate treatment plans are highly important to achieve survival rate that may reach 99% [4]” please add „depending on the type of skin cancer”. Very good prognosis is in BCC but totally diferent situation is in SCC or MM. And in the next paragraph Authors are writting about MM and possible 75% rate of deaths so some inconsistency in thinking is present…
- Response 3:
- The sentence „depending on the type of skin cancer” was added.
- „MM and possible 75% rate of deaths” was explained.
- Point 4: Paragraph from line 85 about NMSC (change to KC) needs revision, read above mentioned article „Detailed head localization and incidence of skin cancers”. In the sentence „It accounts for 30% of all cancers, with a low overall mortality rate despite high occurrence, contrary to MM [8].” It at the beginning of sentence is for BCC, CSCC or NMSC?
- Response 4:
- The paragraph about (KC) has been revised using the suggested article
- „It” at the beginning of the sentence is for KC
- Point 5: In point 4 Correlation between Skin and Blood Metabolites remove „subcutaneous” from skin layers!!!! Subcutaneous tissue is the layer under the skin but totally not the part of it. Skin has 2 layers: epidermis and dermis. I do not understand the connection between this paragraph and skin cancers, the Authors write here about psoriasis, glucose level, but there are no correlations between skin and metabolites in skin cancers?!
- Response 5:
- We thank the reviewer for identifying this error, subcutaneous term was removed as required
- The aim of this subheading was to show which is more preferred to be used for metabolomics studies of skin diseases in general, blood or skin samples. So, to be clearer, the subheading title was changed to „Skin or blood metabolomic profiling for diagnosing and treating skin conditions”
- Point 6: Similar situation in in points 6, 7, and 8. Maybe points from 4 to 8 should be presented together as the examples and as some kind of introduction to similar connection in skin cancers. Think of reorganizing.
- Response 6:
- Points were reorganized by combining some subheadings and forming other new subheadings (Please refer to manuscript and follow the track and change mode to check the modifications that have been done)
- Point 7: What is the take home message from your article especially for clinicians? In what way presented knowledge can be „translated” to management in patients with skin cancer?
- Response 7: We have added a new subheading under the title of new paragraph Clinical implications of the metabolic profile of skin cancer (Page: 23, Lines: 928-935). includes our take home message from our article for clinicians
Reviewer 2 Report
Dear authors,
This review summarizes the contribution of metabolomics technology for evaluating the metabolic profile of human skin cancer via a variety of analytical platforms. Nowadays, there is an urgent need to identify, characterize, and validate informative biomarkers, as well as their clinical implications in every type of cancer including melanoma and other skin cancers. Moreover, the correlation between metabolomics -including microbiota- and treatment, such as immunotherapy and targeted treatment, is of paramount importance in the light of personalised medicine implementation. In general, this review could be considered successful, but the presence of a number of critical shortcomings does not allow recommending it for publication in its current form.
1. A review should not be just a compilation of individual relevant data and the description of various techniques, but the author's view of this scientific field. Authors should have also mentioned potential clinical implications of the metabolic profile of skin cancer in further detail. The section regarding immunotherapy-related biomarkers needs to be further elaborated, too. Challenges in metabolomics that include the development of novel biomarkers evaluating the response to immunotherapy should have been further discussed.
2. The emerging role of microbiota and adipocyte-derived lipids should have been further described. In general, the reference list could be further enriched:
a. Zhang et al. Adipocyte-Derived Lipids Mediate Melanoma Progression via FATP proteins. Cancer Discov. 2018
https://pubmed.ncbi.nlm.nih.gov/29903879/
b. Davar et al. Fecal microbiota transplant overcomes resistance to anti-PD-1 therapy in melanoma patients. Science. 2021
https://pubmed.ncbi.nlm.nih.gov/33542131/
c. Bui et al. Unbiased Microbiome and Metabolomic Profiling of Fecal Samples from Patients with Melanoma. Methods Mol Biol. 2021
https://pubmed.ncbi.nlm.nih.gov/33704734/
d. Frankel et al. Metagenomic Shotgun Sequencing and Unbiased Metabolomic Profiling Identify Specific Human Gut Microbiota and Metabolites Associated with Immune Checkpoint Therapy Efficacy in Melanoma Patients. Neoplasia. 2017
https://pubmed.ncbi.nlm.nih.gov/28923537/
e. Kim et al. Discovery of potential biomarkers in human melanoma cells with different metastatic potential by metabolic and lipidomic profiling. Sci Rep. 2017
https://pubmed.ncbi.nlm.nih.gov/28821754/
3. The section of introduction should be thoroughly revised and shortened. General definitions of cancer and skin could be omitted.
4. There is no information on the process of data collection and the type of review. Authors should have mentioned briefly if, for instance, they followed the PRISMA guidelines for data collection and analysis.
5. The english language and style must be corrected. For example:
line 68: Delete "a" from "a rising evidence"
line 77: "However" instead of "although"
line 141: Delete "into"
line 229: Substitute or delete "great"
line 253 and 256 "Also" could be substituted by "Furthermore" at the beggining of sentences.
Author Response
- Point 1: A review should not be just a compilation of individual relevant data and the description of various techniques, but the author's view of this scientific field. Authors should have also mentioned potential clinical implications of the metabolic profile of skin cancer in further detail. The section regarding immunotherapy-related biomarkers needs to be further elaborated, too. Challenges in metabolomics that include the development of novel biomarkers evaluating the response to immunotherapy should have been further discussed.
- Response 1:
- We thank the reviewer for the queries and suggestions. Therefore, we have added a new paragraph Clinical implications of the metabolic profile of skin cancer
- (Page: 23, Lines: 914-954) with additional references detailing more the clinical implications of the metabolic profile of skin cancer, immunotherapy-related biomarkers, challenges in metabolomics including biomarkers evaluation of immunotherapy response and steps toward precision medicine.
- Point 2: The emerging role of microbiota and adipocyte-derived lipids should have been further described. In general, the reference list could be further enriched:
- Response 2: This has been done as requested. We have added a new paragraph 2 The emerging role of microbiota and adipocyte-derived lipids (Page: 6 and 7, Lines: 226-267) explaining more the important role of microbiota and adipocyte-derived lipids using the suggested references.
- Point 3: There is no information on the process of data collection and the type of review. Authors should have mentioned briefly if, for instance, they followed the PRISMA guidelines for data collection and analysis.
- Response 3: We thank the reviewer for this comment. We did not follow PRISMA guidelines that suits only with systematic review with a specific review design including methods with discussion and results section. Our review type is a literature review that summarize a literature on a general topic not a systematic review. We have utilized PubMed and google scholar for data collection without any use for analytical method to collect and analyse secondary data and without a specific hypothesis or question to be answered.
- Point 4: The English language and style must be corrected. For example:
- line 68: Delete "a" from "a rising evidence"
- line 77: "However" instead of "although"
- line 141: Delete "into"
- line 229: Substitute or delete "great"
- line 253 and 256 "Also" could be substituted by "Furthermore" at the beginning of sentences.
- Response 4: We thank the reviewer for noting this. All were done as requested.
Round 2
Reviewer 1 Report
Authors introduced all suggested remarks.
Thank you.
Reviewer 2 Report
Dear authors,
A significant improvement can be detected in your manuscript. However, some minor revisions are still needed.
1) Authors should have mentioned briefly that this manuscript constitutes a literature review and its data collection process has not been based on any analytical method.
2) The section of clinical implications consists of a general approach towards the benefits of precision medicine to cancer patients. It should be further elaborated.
3) English language and style revisions are needed. Some examples:
Line 70: You mention "However, there is rising evidence that microbiota may also contribute." This sentence seems to be incomplete. It could be rewritten as: "However, there is rising evidence that microbiota may also contribute to skin cancers pathogenesis."
Line 239: Substitute "regarding" for "for"
Line 252: ICT abbreviation should have been explained here instead of line 254.
Line 976: Substitute "response to" for "response in"
Lines 977-992: This sentence needs to be further clarified.
After eliminating at least these remarks, we can recommend this section for publication.
Kind regards,
Author Response
Comments 1: Authors should have mentioned briefly that this manuscript constitutes a literature review and its data collection process has not been based on any analytical method.
Response 1: We thank the reviewer for this comment, this was mentioned briefly in the abstract section
Comment 2: The section of clinical implications consists of a general approach towards the benefits of precision medicine to cancer patients. It should be further elaborated.
Response 2: this section has been further elaborated as required
Comment 3: English language and style revisions are needed. Some examples:
Line 70: You mention "However, there is rising evidence that microbiota may also contribute." This sentence seems to be incomplete. It could be rewritten as: "However, there is rising evidence that microbiota may also contribute to skin cancers pathogenesis."
Line 239: Substitute "regarding" for "for"
Line 252: ICT abbreviation should have been explained here instead of line 254.
Line 976: Substitute "response to" for "response in"
Lines 977-992: This sentence needs to be further clarified.
Response 3: We thank the reviewer for noting this. All were done as requested.